# Understanding General Practitioners’ Antibiotic Prescribing Decisions in Out-of-Hours Primary Care: A Video-Elicitation Interview Study

**DOI:** 10.3390/antibiotics9030115

**Published:** 2020-03-07

**Authors:** Annelies Colliers, Samuel Coenen, Katrien Bombeke, Roy Remmen, Hilde Philips, Sibyl Anthierens

**Affiliations:** 1Centre for General Practice and Skills Lab, Department of Primary and Interdisciplinary Care (ELIZA), Faculty of Medicine and Health Sciences, University of Antwerp, B-2610 Antwerp, Belgium; samuel.coenen@uantwerpen.be (S.C.); katrien.bombeke@uantwerpen.be (K.B.); roy.remmen@uantwerpen.be (R.R.); hilde.philips@uantwerpen.be (H.P.); sibyl.anthierens@uantwerpen.be (S.A.); 2Department of Epidemiology and Social Medicine (ESOC), Faculty of Medicine and Health Sciences, University of Antwerp, Universiteitsplein 1, B-2610 Antwerp, Belgium; 3Vaccine & Infectious Disease Institute (VAXINFECTIO), Faculty of Medicine and Health Sciences, University of Antwerp, Universiteitsplein 1, B-2610 Antwerp, Belgium

**Keywords:** antibiotics, video observation, elicitation interview, out-of-hours care, decision-making, general practitioners

## Abstract

Infections are the most common reason why patients consult out-of-hours (OOH) primary care. Too often there is an overprescribing of antibiotics for self-limiting infections and general practitioners (GPs) do not always choose the guideline recommended antibiotics. To improve antibiotic prescribing quality, a better understanding is needed of the (non) antibiotic prescribing decisions of GPs. This study sets out to unravel GPs’ (non) antibiotic prescribing decisions in OOH primary care. We video-recorded 160 consultations on infections during OOH primary care by 21 GPs and performed video-elicitation interviews with each GP. GPs reflected on their decision-making process and communication while watching their consultation. A qualitative thematic analysis was used. GPs found that their (non) antibiotic prescribing decision-making was not only based on objective arguments, but also subconsciously influenced by their own interpretation of information. Often GPs made assumptions (about for example the patients’ reason for encounter or expectations for antibiotics) without objectifying or verifying this with the patient. From the beginning of the consultation GPs follow a dichotomous thinking process: urgent versus not urgent, viral versus bacterial, antibiotics versus no antibiotics. Safety-netting is an important but difficult tool in the OOH care context, with no long-term follow-up or relationship with the patient. GPs talk about strategies they use to talk about diagnostic uncertainty, what patients can expect or should do when things do not improve and the difficulties they encounter while doing this. This video- elicitation interview study provides actionable insights in GPs’ (non) antibiotic prescribing decisions during OOH consultations on infections.

## 1. Introduction

Improving the quality of antibiotic prescribing is essential to overcome the global health problem of antibiotic resistance. General practitioners (GPs) too often prescribe antibiotic treatment and in addition they often choose not-guideline recommended antibiotics [1]. Out-of-hours (OOH) care is particularly relevant in this respect [2]. During OOH GPs are mostly confronted with patients with infections [3], and antibiotic prescribing quality shows room for improvement [4,5,6,7]. In previous research we learned that GPs feel that the OOH context influences their prescribing behaviour, because of their different professional role, first and only contact with this patient, time pressure, etc [8,9]. What still remains unclear is how these beliefs influence their decision-making process and communication with patients.

In order to get a clear picture of why and how GPs make antibiotic prescribing decisions, this ‘picture’ should reflect GPs’ real life practice. Video-recording primary care consultations is a valuable method increasingly used to provide rich data on actual patient–doctor interactions, [10,11] but has not yet been used in out-of-hours (OOH) primary care [12]. Video-elicitation interviews, depending on their epistemological view also called reflective video-stimulated interviews, video-stimulated recall, video reviews or video-cued narrative reflection, is a qualitative method particularly well suited for investigating doctor-patient interactions. During an interview with the participant the researcher uses a recent video-recorded clinical interaction as an elicitation tool. The video is used as a prompt to explore the interviewee’s thoughts, beliefs and feelings [13,14]. It provides insight into what actually happens inside the consultation room and in GPs’ minds during a consultation, shedding light on the underlying decision-making process [15].

Therefore, we set out to unravel GPs’ decision-making during consultations on infections in OOH primary care through video-elicitation interviews, with a special focus on (non) antibiotic prescribing. A video of their consultation was used as a prompt to explore tacit knowledge (that what they know but find difficult to describe) [16], beliefs, attitudes, social influences, communication, etc. that drove their behavior [17]. To improve antibiotic prescribing quality in this specific setting, better understanding of GPs’ beliefs in action on their antibiotic prescribing decision-making is crucial to inform the development of effective interventions, create awareness and guide changes in attitudes and practices.

## 2. Methods and Materials

### 2.1. Study Context

In Belgium OOH primary care provides 24-h-care by GPs during weekends and bank holidays for acute problems. Belgian OOH care is mostly organised in large-scale general practitioner cooperatives (GPCs). GPs see mostly patients they have never met before; for follow-up, patients are sent back to their regular GP. There is an electronic medical health record with very limited (and often no) patient information, there is a high work load and there is no direct access to diagnostic tests. There is no triage system yet in most GPCs, patients are free to consult the GPC. There is a fee-for-service system, but consultation fees are largely reimbursed by the compulsory public insurance system. For a consultation in OOH care during daytime, the out of pocket payment for patients is between one and six euros, depending on their social care status. There is a possibility to use a third-party payment if the patient is in financial need (Figure 1).

This study was conducted at the central Antwerp GPC in Belgium. It covers a population of 187,000 inhabitants with a diverse ethnic background. All 175 GPs of the region have the obligation to participate in the OOH care system.

This study is part of the BAbAR (Better Antibiotic prescribing through Action Research) project, that uses a participatory action research (PAR) approach, with the goal to improve the quality of antibiotic prescribing of GPs in OOH care [2].

### 2.2. Organisation of the Video-Observations

As part of BAbAR, the study was co-created with the stakeholders from the GPC (GPs, manager, board). The videos were recorded with a small web camera placed above the computer screen. GPs could start and stop the camera themselves. More information on the how and why choices were made in the set-up of the study has been provided in another paper [12].

### 2.3. Study Design

We performed qualitative elicitation interviews after video-recording GP consultations on infections at a GPC. During the study period from August 2018 until November 2018, we used a convenience sample out of the GPs on call. Within this sample we purposively selected GPs to reflect the variety of GPs at the GPC (sex/age). In total, 160 videos were recorded from 21 participating GPs. In total, 31 GPs were asked to participate. Seven GPs who were invited, refused to be video-recorded, mostly because of personal reasons (“It gives me too much stress to be filmed”, “I’m not feeling well at the moment”,…) and three GPs could not be reached by email or telephone.

In order to stimulate safe and thus more authentic reflections in the elicitation interviews, every participating GP selected one of their own recorded consultations. Then the videos were used as prompts to reflect back on their decision-making process and its effect on communication within OOH consultations on infections. The chosen video was discussed during a face-to-face elicitation interview that took place in their own practice, within two weeks after the recording. Many participants knew the interviewer as a former colleague, as she used to work as a GP in this region. The interviews followed a semi-structured interview guide (Appendix A). The guide was inspired by another elicitation interview study on antibiotics and an article on video-elicitation methods, and further developed by two of the authors (AC and SA) [13,15]. However, the interviews were mostly guided by what happened in the video-recorded consultations and the interviewee’s responses. Data collection and data analysis was an iterative process, and the interview guide was adapted along this process.

During the interviews, the interviewer and interviewees paused the video at key points (end of the patients problem presentation, before and after clinical examination, after the diagnosis was discussed, …) or whenever they felt it was relevant to reflect on the decision-making process or communication.

All interviews were audio-recorded and transcribed verbatim.

### 2.4. Data Analysis

Data analysis occurred alongside and informed data collection. As new data were collected the researcher(s) read each new transcript and re-explored earlier transcripts.

The interview transcripts were analysed systematically following the steps of the thematic analysis method [18,19].

In a first step, data familiarization was done reading through all the transcripts and viewing the corresponding video-recorded consultation while making memos by the main researcher who was also the interviewer (AC). The first three interviews were open coded by two researchers independently: AC, a GP and SA, a medical sociologist. Together they identified the first relevant themes from this coding process. They used an inductive approach driven by the data. The next two interviews were analysed following the same approach, now by AC and KB, a GP/communication skills trainer. The different themes were revised toward a general thematic framework. The themes were chosen because of their recurrence within interviews, their relevance for our research question, and/or their innovativeness. All interviews were then further analysed by AC during an iterative process of moving forward and backward between the data and the thematic analysis. During two data discussion sessions (one after the analysis of the first five interviews and one after the first 17 interviews), the themes were reviewed, discussed and defined by the wider team: SC, MD/researcher on infectious diseases in primary care and HP, GP/researcher on OOH primary care. We discussed which themes were relevant for the research question and how these should be interpreted and presented and if the themes represented our data sufficiently. This researcher triangulation was performed to enhance trustworthiness.

We strived for an in-depth analysis. So rather than just describing what was said, we also searched for interpretation and explanation of how participants understand and give meaning to what happened inside the consultation room, focusing on the (non) antibiotic prescribing decisions.

### 2.5. Ethics

The study was approved by the Ethics Committee of the Antwerp University Hospital/University of Antwerp (reference number 17/08/089), and registered at clinicaltrials.gov (NCT03082521). Permission for the video-recordings was obtained from the Belgian Committee of Health of the Commission for the Protection of Privacy (SCSZG/18/067).

## 3. Results

Twenty-one GPs participated. Table 1 shows their characteristics.

Relevant quotes are provided as Appendix A.

We identified three themes (Figure 1).

(1)Influence of unverified interpretation of information or assumptions(2)Dichotomous thinking and communication(3)Safety-netting: strategies and difficulties

### 3.1. Theme 1: Influence of Unverified Interpretation of Information or Assumptions

While reviewing their consultations and reflecting on what was noticed, GPs became aware that not only objective criteria influenced their decision-making, but also experience, intuition, interpretations, estimations and assumptions.

#### 3.1.1. Exploring Ideas, Concerns and Expectations

The communication tool of probing ideas, concerns and expectations (ICE) of patients to explore their reasons for encounter [20] was known and accepted among the participating GPs. However, few of them actively applied this method, but instead assumed why the patient came to the OOH GP. Nonetheless, these assumptions determined their decisions. Whilst watching their own consultation, they concluded that it could have helped them in their decision-making if they would have explored the patient’s perspective more elaborately. For example, they acknowledged that knowing if the patient expected antibiotics or not, or if they rather sought reassurance or advice would have helped them tailor their communication in a patient-centred way. Often, GPs felt pressured to prescribe antibiotics, also without really exploring whether the patient actually wanted antibiotics or discussing this with the patient.

#### 3.1.2. Patient’s Presentation & GP’s Interpretation

Rather than its content, details of how and what patients said and the way they presented their problem, influenced the thinking process of GPs to a great extent. Often patients tended to present their problem as “doctorable” to legitimize their OOH visit [21]. The emphasis or dramatization the patients put on their medical concern was interpreted by the GP as it being more urgent, more severe or as a request for antibiotics. Typically in OOH care, patients are unknown to the GPs, so they could not fall back on previous experiences with this patient. The way patients told their story, their job/background, the overall first impression, their ethnicity, all played a role somehow in how the GPs interpreted the problem. They used these small elements to assess the patients’ help seeking behaviour, the patients’ ability to assess the need to see a GP during OOH, the patients’ expectations (on for example antibiotics), and so on.

Together with more objective anamnestic and clinical examination findings, the illustrated unverified interpretation of information or assumptions were subconsciously influencing the GPs decision-making process and guided their consultation and management of the infection.

### 3.2. Theme 2: Dichotomous Thinking and Communication

Working at the GPC is different from working in your own GP practice, and this influenced GPs’ consultation behaviour, decision-making and communication. They all confirmed that the recorded consultation did reflect their consultation style and that they were not a completely different doctor at the GPC. However, GPs felt their duty is different at the GPC. They tried to limit consultation time. Therefore, their history taking was more superficial, ignoring patient cues more often than they would normally do. Their main focus was on ruling out dangerous and urgent pathologies. Their thinking process was already attuned to two dichotomies from the beginning of the consultation. First, they tried to rule out dangerous and urgent conditions by assessing alarm symptoms. Second, their thinking process was focused on distinguishing between viral or bacterial infection. Only when thinking fast and dichotomous did not fit the full clinical picture, no clear pattern was recognized or assumed, they dug deeper in their clinical thinking.

Consequently, their thinking process was already focused from the start of the consultation into: ‘should I or should I not prescribe antibiotics’.

This dichotomous thinking was also reflected and supported in the communication with the patient and GP’s use of language. **GPs** explained to the patient they had a viral infection, first to educate patients, but also to counter any expectations for antibiotics, without asking if there were expectations for antibiotics. They used power words such as ‘resistance’, ‘a powerful drug’, ‘conquering an infection’, and so on to be persuasive, and to strengthen their message to nudge the patient to take action accordingly or to withhold antibiotics, but also to show they acknowledged the severity of the patient’s symptoms. However, they acknowledged whilst reflecting on their videos that it might have been more appropriate to explain in lay language to the patients what they meant.

There are GPs who felt very confident sending the patient home with self-care advice and home remedies, when the decision was made not to prescribe antibiotics. But there were others who preferred to prescribe a list of symptomatic medications, again to acknowledge the severity of patient’s symptoms and to meet the patients alleged expectations.

### 3.3. Theme 3: Safety-Netting: Strategies and Difficulties

All interviewees discussed elements of safety-netting with their patients.

GPs often chose a ‘watchful waiting’ strategy and explained alarm symptoms and the expected evolution. Yet, they encountered difficulties doing this in the OOH setting. With non-native speakers they felt uncertain if their message was understood clearly. Also, as mentioned earlier, GPs on call mostly treat unfamiliar/new/unknown patients, consequently the assessment of these patients’ presentation of symptom burden, help seeking behaviour and level of anxiety was complicated. Some patients did not have a regular GP. Within the region there are many GPs that do not take in new patients or the waiting time to see a GP is very long. All these elements complicated the decision whether or not the “wait and see” strategy was a safe option.

They used communication strategies to support this safety-netting such as repeating the message or searching for confirmation with the patient, providing information about alarm symptoms, and giving a clear recommendation on future help seeking behaviour. Interestingly, they did not only intensify this strategy when they felt the patient had not understood the message very well, but also when they felt uncertain themselves about for example the diagnosis or prognosis. They found it difficult to communicate this uncertainty.

Some GPs used delayed prescribing as a safety-netting tool. Others felt that it was more important to refer them back to their regular GP and they used safety-netting as a bridge to ensure continuity of care, but they acknowledged that it might be difficult for the regular GP to re-evaluate something they did not see at the first presentation.

The focus was on good clinical work, but not necessarily on building a long-term trusting relationship. They were aware that the patient they were helping is a patient of one of their colleagues, to whom they wanted to be loyal and correct. GPs wanted to give the best possible care to make a professional impression on their colleagues. Sometimes prescribing an antibiotic was used as an umbrella technique. Being acquainted with the GP of the patient influenced their decision. For example, if getting in touch with their regular GP was considered difficult for the patient, this could lower the threshold to prescribe an antibiotic, but otherwise it could convince and reassure them to use the ‘wait and see’ strategy. Again, they made assumptions on how the regular GP would act and how accessible that GP would be in the next week.

## 4. Discussion

### 4.1. Main Findings

In this study, using video-recorded GP consultations on infection in OOH primary care, we elicited several actionable insights in GPs’ (non) antibiotic prescribing decisions. While GPs are trained to base their decisions on objective findings, reflecting on their consultation they noticed that their decisions are influenced by assumptions and interpretations for example about the patients’ reasons to visit the GPC, their ICE, safety-netting abilities, illness behaviour, etc., without verifying them. GPs first rule out what is urgent and second dichotomise in ‘should I prescribe an antibiotic or not’, rather than assess why the patient comes in to see a GP during out of hours. GPs acknowledge the importance of safety-netting, particularly in the OOH setting, but this OOH context complicates their safety-netting abilities and therefore GPs do not always feel confident enough to use the “wait and see” approach.

As described by Kahneman: fast thinking and unconscious complex decision-making, such as GPs do during a consultation, is not always appropriate nor correct [22]. GPs often work on autopilot and systematically pay attention to some information more than others [16]. Clinical reasoning also involves integrating intuitive and analytical thinking processes [23]. It was only while watching their consultation that GPs in our study became aware of the subconscious aspects of their thinking and how these guided the consultation and their prescribing decision. These aspects were real blind spots for these GPs. They often interpreted the reason why a patient comes in to see the GP. Other studies confirm that GPs seldom make patient expectations explicit [24,25], although exploring ICE has been shown to lead to fewer new medication prescriptions [20] and training GPs in eliciting the expectations for antibiotics or any concerns or worries has been successful in reducing antibiotic prescribing for respiratory tract infections [26,27,28].

What the patients says and how they bring their problem presentation influenced the GPs’ thinking process. Doctorability is the tendency of patients to present their medical concern as “doctorable” to legitimize their visit [21]. At the GPC, a patient might be more inclined to present themselves as “doctorable”, and this could affect GPs’ thinking process and the feeling of pressure to “help” the patient (often perceived by GPs as a request for an effective treatment such as an antibiotic or lots of symptomatic medications). The approach of the GPs in our study was very much focused on delivering a diagnosis and treatment in limited time. From problem presentation to management their thinking was focused on urgent vs non urgent, viral vs. bacterial and should I prescribe antibiotics or not? The GP on call has a specific professional role, which is ruling out the urgent and treating what is necessary. The specific goal the GP sets out for him/herself determines the general direction of the communication [29]. Cabral et al. showed in their study on parent–clinician communication on antibiotic prescribing for children with respiratory tract infections that the most common problem presentation was “symptoms only”, which implied parents were seeking medical evaluation, rather than an exact diagnosis or particular treatment [30]. An intervention from Van Uum et al. aimed at optimising pain management in childhood acute otitis media changed GPs perceptions, from treating the infection with antibiotics to treating symptoms [31].

Safety-netting includes the communication of uncertainty and plans for follow-up [32]. Patients in OOH present with acute problems and can deteriorate quickly, all the more reason why safety-netting is an important part of the consultation. But the lack of a trustworthy long term doctor–patient relationship, being unfamiliar with the usual help seeking behaviour of unknown patients, limited diagnostic and follow-up options, language difficulties, loyalty to the patients’ own GP, etc., makes safety-netting more challenging. Continuity of care and a trusting doctor–patient relationship have been shown to be key elements for acceptance of antibiotic prescribing decisions for children with RTIs [33], which is lacking in OOH care.

### 4.2. Strengths and Weaknesses/Limitations

Our work is limited by some of our design decisions. This study was conducted at a single OOH centre. The perspectives we describe also reflect consultations at a particular point in time and in a particular setting. There could be differences between GPs who accept to be video-recorded and the ones who refuse to participate. And because of the time delay between the video-recording and the interview there could be some alterations in the recall of why certain prescribing decisions were made. Although accurate recall of cognitive process could have been altered, the reflection on the behaviour elicited in the moment of interviewing are equally important [17].

A major strength of this study was the rich data that was collected using the video-elicitation interview method. It allowed us to look at real GP behaviour and to retrospectively reflect on it with the participating GPs and uncover unconscious drivers behind made choices. We strived for a reflective approach, which aims for reflection on and interpretation of behaviour as the participant understands it. 

The interviewer is a former colleague of the participants and works in research now; thus, has both an insider and an outsider role in this research setting [34]. Participants are more open and more likely to reflect on their emotions when interviewed by a peer [35]. A reflexive journal was kept by the interviewer. The finding of the dichotomous thinking process could be enhanced because of the ongoing overall study on reducing antibiotic prescribing. A safe environment for GPs was created by repeatedly stating that for this part of the study the goal was to learn, a nonjudgmental approach was used and GPs selected the video themselves.

### 4.3. Implications for Practice and Further Research

Continuous training, guidelines and interventions on antibiotic prescribing mostly focus on rational criteria to make a prescribing decision. We learned that decisions are largely influenced by subconscious interpretation of why and how a patient presents himself, their safety-netting capabilities and the OOH context. Communication training has been proven successful to reduce inappropriate prescribing [27,28]. Training GPs on awareness of unconscious elements [16,36] or on communication of diagnostic reasoning and uncertainty could be an added value and focus for further research. In addition, training patients to make their ICE explicit when visiting the GP on call and providing specific tools for safety-netting during OOH, could be very useful and the effect on antibiotic prescribing should be further studied. A more in-depth analysis of the video-recorded consultations could bring more insight in how communication plays a role in antibiotic prescribing decisions and results could help to develop communication training and clear recommendations for GPs.

## 5. Conclusions

This video-elicitation interview study contributes to a better understanding of GPs’ (non) antibiotic prescribing decisions during in OOH consultations on infections and elicited actionable insights in consultation behaviour and communication strategies. By reflecting on what actually happens during a consultation, GPs became aware of the many unverified interpretations and assumptions they make. They use a dichotomous fast thinking approach and communication to diagnose a viral or a bacterial infection throughout the consultation and consequently to prescribe an antibiotic or not. Furthermore, we achieved a better understanding of how GPs use safety-netting and its difficulties within the OOH context. Video-elicitation interviews could not only be used as a data collection tool for research, but also as an intervention itself as continuous medical education to enhance the quality of antibiotic prescribing, since it unravels the subconscious elements that drive GPs’ prescribing decisions.

## Figures and Tables

**Figure 1 antibiotics-09-00115-f001:**
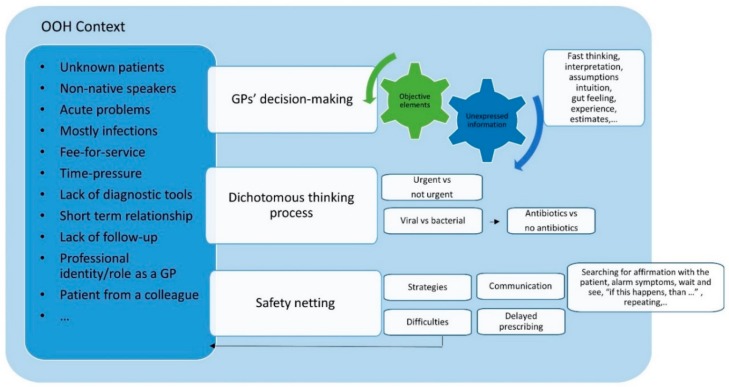
Elements of general practitioners’ (GPs’) antibiotic prescribing decisions during out-of-hours (OOH) primary care consultation on infections.

**Table 1 antibiotics-09-00115-t001:** Characteristics of the general practitioners (GPs) interviewed.

Number of Participating GPs	21
Age in years Mean (SD) Median Range (min-max)	43.1 (13.69) 39 26-64
Years in practice Mean (SD) Median Range (min-max)	15.7 (12.96) 12 1-38
Gender distribution Male Female	8 13
Type of GP practice they work in during regular office hours (outside OOH care) Solo Duo Group Community health centre	2 1 17 1
GP trainee (GP in specialty training)	2
Duration of the interviews in minutes Mean (SD) Range (min-max)	43′ (35′-50′) 34′-57′

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
