# Peer review of "Understanding General Practitioners’ Antibiotic Prescribing Decisions in Out-of-Hours Primary Care: A Video-Elicitation Interview Study"

_antibiotics, 2020, doi:10.3390/antibiotics9030115_

Round 1

Reviewer 1 Report

Review report on manuscript of Understanding general practitioners’ antibiotic prescribing decisions in out-of -hours primary care: a video-elicitation interview study

The authors provide a very interesting manuscript, which use video-elicitation interview method to describe how GPs made antibiotic prescribing decisions in the OOH context. The research method is valuable and the sample size (21 GPs) is ok for a qualitative study. Further, it contributes novel descriptive data to a relevant less-researched area. The paper is well presented, and English language and style are fine.

Border comments

Relied on stimulus, the video elicitation interview is a very sensible way to prompt GPs to discuss their (non) antibiotic prescribing decisions in greater detail. However, in the Introduction Section, it can be more helpful to provide a brief description on the concept or method of video elicitation interview, which can help readers to better understand how this method works, than providing its other names.

In the Method Section, please (i) provide a description on OOH context in ‘2.1 Study context’, although some characteristics of OOH context has been given in Figure 1. Also, please (ii) clarify below information in ‘2.2 Organisation of the video-observations’:

  • How many GPs were invited? It was noted there were 21 GPs participated, 7 GPs refused and 3 GPs who could not be reached. So, were there in total 31 GPs invited, and what was the selection criteria?
  • How were the 160 videos recorded and by whom?

If answers for query (i) and query (ii) have been provided in another paper, please give a brief description here (i.e. one or two sentences) and clearly refer that paper.

(iii) In ‘2.3 Study design’, it is a bit confusing of the description on selected GPs (line 86-88). Are they the same 21 GPs? If yes, please describe the recruitment process in a more continuous way.

The Result Section is very interesting with three themes identified. Please provide supplementary information on how the themes generated, including subthemes, categories, or codes particularly for theme 2 and theme 3.

Specific comments

  1. In Table 1, the ‘Type of practice during of office hours’ needs to be clarified, noting it is GPC office hours or OOH office hours.
  2. In Table 1, the meaning of ‘General practitioners in professional training’ is not clear. Please clarify what it means.

Author Response

Dear reviewer,

thank you for your time and effort to revise our manuscript.

We are happy to respond to the points and concerns that were brought up.

We will go through the remarks one by one here below.

The authors provide a very interesting manuscript, which use video-elicitation interview method to describe how GPs made antibiotic prescribing decisions in the OOH context. The research method is valuable and the sample size (21 GPs) is ok for a qualitative study. Further, it contributes novel descriptive data to a relevant less-researched area. The paper is well presented, and English language and style are fine.

Border comments

Relied on stimulus, the video elicitation interview is a very sensible way to prompt GPs to discuss their (non) antibiotic prescribing decisions in greater detail. However, in the Introduction Section, it can be more helpful to provide a brief description on the concept or method of video elicitation interview, which can help readers to better understand how this method works, than providing its other names.

* Thank you for this comment. We elaborated on the method in the introduction section and we added: “During an interview with the participant the researcher uses a recent video-recorded clinical interaction as an elicitation tool. The video is used as a prompt to explore the interviewee’s thoughts, beliefs and feelings.”

In the Method Section, please (i) provide a description on OOH context in ‘2.1 Study context’, although some characteristics of OOH context has been given in Figure 1.

*Thank you for this comment. We added following phrases in the study context section: “In Belgium OOH-primary care provides 24/24h-care by general practitioners during weekends and bank holidays for acute problems. Belgian OOH care is mostly organised in large-scale general practitioners cooperatives (GPCs). GPs see mostly patients they have never met before, for follow-up there are being sent back to their regular GP. There is an electronic medical health record with very limited information (and often none), there is a high work load and there is no direct access to diagnostic tests. There is no triage system yet in most GPCs, patients are free to consult the GPC. There is a fee-for service system, but consultation fees are largely reimbursed by the compulsory public insurance system. The out of pocket payment for patients is between one and six euros, depending on their social care status, for a consultation in OOH care during daytime. There is a possibility to use a third party payment if the patient is in financial need.”

Also, please (ii) clarify below information in ‘2.2 Organisation of the video-observations’:

  • How many GPs were invited? It was noted there were 21 GPs participated, 7 GPs refused and 3 GPs who could not be reached. So, were there in total 31 GPs invited, and what was the selection criteria?

*Indeed. There were 31 GPs invited to participate. For this we used a convenience sample out of the GPs who were on call during the weekends the study was running. We chose the ones who were responsible for the consultations (so not the home visits). Within this sample we purposively selected GPs to have  a variation in age and sex. This is explained in 2.3 Study design. Information about the participating GPs was replaced to 2.3 to enhance clarity about the recruitment process. (see also further) We also added in 2.3 study design: “In total 31 GPs were asked to participate.”

  • How were the 160 videos recorded and by whom?

If answers for query (i) and query (ii) have been provided in another paper, please give a brief description here (i.e. one or two sentences) and clearly refer that paper.

*In 2.2 Organisation of the video-observations we added: “The videos were recorded with a small web camera placed above the computer screen. GPs could start and stop the camera themselves. More information on the how and why choices were made in the set-up of the study has been provided in another paper. (Colliers et al.  Looking Inside the Out-of-Hours Primary Care Consultation: General Practitioners’ and Researchers’ Experiences of Using Video Observations as a Method. International Journal of Qualitative Methods 2019, 18)”

(iii) In ‘2.3 Study design’, it is a bit confusing of the description on selected GPs (line 86-88). Are they the same 21 GPs? If yes, please describe the recruitment process in a more continuous way.

*Thank you for this comment. To enhance clarity the information about the recruitment and participating GPs is now put together in 2.3 study design.

The Result Section is very interesting with three themes identified. Please provide supplementary information on how the themes generated, including subthemes, categories, or codes particularly for theme 2 and theme 3.

*A step by step thematic analysis approach was used to generate the different themes and subthemes. We started with an open coding in an inductive way. Within the codes patterns were sought and brought together as themes. And from that a general thematic framework was constructed. The themes were chosen because of their recurrence within all interviews, their relevance for our research question, and their innovativeness. The team organized several data sessions to discuss about which themes were relevant for our research question and how these should be interpreted and presented and to see if the themes represented our data sufficiently. We added in 2.4 Data analysis: “The themes were chosen because of their recurrence within all interviews, their relevance for our research question, and/or their innovativeness. And “We discussed which themes were relevant for the research question and how these should be interpreted and presented and if the themes represented our data sufficiently.”

Specific comments

  1. In Table 1, the ‘Type of practice during of office hours’ needs to be clarified, noting it is GPC office hours or OOH office hours.

*Thank you for this comment. We changed this to: “Type of GP practice they work in during regular office hours (outside OOH care)”

  1. In Table 1, the meaning of ‘General practitioners in professional training’ is not clear. Please clarify what it means.

*Thank you for this comment. In Belgium there is a two years training after finishing medical school were you combine working as a GP under supervision and some education. We changed it to: “GP trainee (GP in Specialty Training)”

Reviewer 2 Report

This is an interesting contribution to our understanding of how GP prescribing behaviour. I believe it is worth publishing but it would add greatly to the quality of the paper if results could include some statistics or even just descriptions of tendencies.  Also, the recommendations from the study should be fleshed out. Limitations of the study also need to be described -- several potential limitations are missing.  The section on limitations should not be so focussed on strengths.  You could describe the strengths but then describe the weaknesses in full also. Finally, it would be useful to propose follow-on questions for further research.  Indeed this project should be seen as a stepping stone towards future, more in-depth work. 

Other comments/concerns:

Why focus on (non) prescribing decisions?

The first sentence in the Abstract doesn’t make sense.  “Infections are the number one reason to consult out-of-hours (OOH) primary care with overprescribing of antibiotics.”

“ not objectified interpretation of information or assumptions.” Could you explain?

“ safety netting strategies and difficulties were highlighted” Not very clear.

“There is a fee-for service system, but consultation fees are largely reimbursed by the compulsory public insurance system.” Having to pay any money out of pocket initially can be a barrier to consulting a doctor, even if the fee is reimbursed later.  You may want to mention what it costs roughly so that the reader can gauge if this could deter patients (and thereby gauge if there is any risk of selection bias from the cost of treatment).

Having a quarter of the potential participants refuse to be filmed may suggest some selection bias amongst the GPs – no? A possible limitation to mention.

“In order to stimulate safe and thus more authentic reflections in the elicitation interviews, every participating GP selected one of their own recorded consultations.” I don’t understand this. It sounds counter-intuitive (unless they chose at random).

It is good that a time limit was given such that interviews took place within 2 weeks of recording. However, even after one week I would imagine that GPs could start to forget the logic behind their prescribing decision, no?

Was there any follow-up after any more formal diagnosis (e.g. lab results)?

“This researcher triangulation was performed to enhance trustworthiness. “ I think you should find a more suitable word than “trustworthy”  --- perhaps reliability? Consistency? To remove any bias in interpretation?

Did the GPs have any training on AB prescribing (beyond medical school) or were they given any materials on AB prescribing at any point in time?

Findings are interesting but it would be useful to have some statistics (especially within Theme 1) to be able to learn more concretely from them.  Even just “x was more likely to lead to y”.

It would also be useful to have some numbers to understand where the thresholds lie. For example, how long does the expected wait to see a routine (or followup) GP for the OOH GP to feel that she/he should prescribe ABs?

English needs some work, especially in the Abstract.

A few English mistakes in the main text:

Word missing: Video recording of primary care consultations is a valuable

“A video of their own consultation was used as a prompt to explore the tacit knowledge (that what they know but find difficult to describe)[16], beliefs, attitudes, social influences, communication… that drove their behaviour.“

“beliefs-in action” (there should be no hyphen)

“All about 175 GPs of the region have the obligation to participate in the OOH care system.”

“If this happens… than…” (it should be “then”)

Author Response

Dear reviewer,

thank you for your time and effort to revise our manuscript.

We are happy to respond to the points and concerns that were brought up.

We will go through the remarks one by one here below.

This is an interesting contribution to our understanding of how GP prescribing behaviour. I believe it is worth publishing but it would add greatly to the quality of the paper if results could include some statistics or even just descriptions of tendencies.  Also, the recommendations from the study should be fleshed out. Limitations of the study also need to be described -- several potential limitations are missing.  The section on limitations should not be so focussed on strengths.  You could describe the strengths but then describe the weaknesses in full also. Finally, it would be useful to propose follow-on questions for further research.  Indeed this project should be seen as a stepping stone towards future, more in-depth work. 

*Thank you for your comments. Limitations were added to the specific section (see further in the text) . A thorough explanation on why we did not choose to include any statistics could be found further in the text. Follow-on questions for further research were added.

Other comments/concerns:

Why focus on (non) prescribing decisions?

*Thank you for this question. Antibiotics are mostly prescribed in primary care and there is the growing problem of antibiotic resistance. Although there have been for example yearly public campaigns , guidelines, and so on there is still an important overprescribing of antibiotics, although many. To be able to understand why and how GPs make this decision we decided to focus on their prescribing decisions. Knowledge about what influences (non) prescribing decisions can help to tailor future interventions, continuous medical education, (communication) training in medical school, etc. Because of the limited word count we did not elaborate on this in the abstract, however it has been stated in the introduction.

The first sentence in the Abstract doesn’t make sense.  “Infections are the number one reason to consult out-of-hours (OOH) primary care with overprescribing of antibiotics.”

*Thank you for this comment. We changed this sentence in: “Infections are the  most common reason why patients  consult out-of-hours (OOH) primary care. Too often there is an overprescribing of antibiotics for self‐limiting infections and general practitioners (GPs) do not always choose the guideline recommended antibiotics.”

“ not objectified interpretation of information or assumptions.” Could you explain?

*Thank you for this comment. We tried to elaborate on it to make it more clear for the reader. “GPs found that their (non) antibiotic prescribing decision making was not only based on objective arguments, but also subconsciously influenced by the GPs’ own interpretation of information. Often they made assumptions (about for example the reason for encounter, the patients’ expectations for antibiotics,…) without objectifying or verifying this with the patient.

“ safety netting strategies and difficulties were highlighted” Not very clear.

*Thank you for this comment. We elaborated this sentence: “Safety netting is an important but difficult tool, in the OOH care context, with no long-term follow-up or relationship with the patient. GPs talk about strategies they use to talk about diagnostic uncertainty, what patients can expect or should do when things don’t improve and the difficulties they encounter while doing this.”  

“There is a fee-for service system, but consultation fees are largely reimbursed by the compulsory public insurance system.” Having to pay any money out of pocket initially can be a barrier to consulting a doctor, even if the fee is reimbursed later.  You may want to mention what it costs roughly so that the reader can gauge if this could deter patients (and thereby gauge if there is any risk of selection bias from the cost of treatment).

*Thank you for this comment. The barrier in Belgium to consult OOH care is low. We added this to explain: “There is a fee-for service system, but consultation fees are largely reimbursed by the compulsory public insurance system. For a consultation in OOH care during daytime, the out of pocket payment for patients is between one and six euros, depending on their social care status. There is a possibility to use a third party payment if the patient is in financial need.”

Having a quarter of the potential participants refuse to be filmed may suggest some selection bias amongst the GPs – no? A possible limitation to mention.

*Thank you for this comment. Indeed the sampling of the participants, and some refusals to participate could lead to some form of bias. However our goal was not to present generizable data but rather to strive for transferability. We tried to do this with a sufficient varied sample and to describe elaborately the context and participants for the reader to judge de degree of transferability for their own setting and context . We added following sentence to the limitations section. “There could be differences between GPs who accept to be video-recorded and the ones who refuse to participate.”

“In order to stimulate safe and thus more authentic reflections in the elicitation interviews, every participating GP selected one of their own recorded consultations.” I don’t understand this. It sounds counter-intuitive (unless they chose at random).

*Thank you for this question. The stakeholders were involved from the beginning of the setup of the study and it was co-designed with them. An elaborate explorative study was conducted before setting up the study (Colliers et al.  Looking Inside the Out-of-Hours Primary Care Consultation: General Practitioners’ and Researchers’ Experiences of Using Video Observations as a Method. International Journal of Qualitative Methods 2019, 18)) One of the items the stakeholders discussed was that they themselves wanted to select the video to discuss with the researcher. They explained that they chose this so they wouldn’t feel judged or they wouldn’t feel they had to defend their selves. To be able to establish sufficient trust to record the GPs the choice was made they could choose any video that they wanted to discuss with the researcher. Important to know it that all of the 21 GPs chose their most difficult or challenging consultation. They chose the ones they perceived as more difficult (the one they struggled with, were in doubt, didn’t know exactly the diagnosis, and so on) because these were the ones they could learn from themselves by reflecting on it. A more clear reference to the paper was added in 2.2 Organisation of the video-observations: “More information on the how and why choices were made in the set-up of the study has been provided in another paper”

It is good that a time limit was given such that interviews took place within 2 weeks of recording. However, even after one week I would imagine that GPs could start to forget the logic behind their prescribing decision, no?

*Thank you for this comment. Indeed after two weeks an accurate recall of their decision making process could be altered. We strived for a reflective approach, which aims for reflection on and interpretation of behaviour as the participant understands it. But there is certainly overlap with a more recall approach, whereby cognitive processes underlying and taking place during actual behaviour are being elicited That’s why we state “The interviews took place within two weeks after the consultations, accurate recall of cognitive process could have been altered, but the reflection on the behaviour elicited in the moment of interviewing are equally important.” We added to the limitations section: “because of the time delay between the video recording and the interview there could be some alterations in the recall of why certain prescribing decisions were made.”

Was there any follow-up after any more formal diagnosis (e.g. lab results)?

*Thank you for this question. There is no follow-up of the patient after the OOH visit. Some extra information about the OOH context has been provided in 2.1: “GPs see mostly patients they have never met before, for follow-up there are being sent back to their regular GP. There is an electronic medical health record with very limited information (and often none), there is a high work load and there is no direct access to diagnostic tests.”

“This researcher triangulation was performed to enhance trustworthiness. “ I think you should find a more suitable word than “trustworthy”  --- perhaps reliability? Consistency? To remove any bias in interpretation?

*Thank you for this comment. The word ‘trustworthiness’ is one of the elements of scientific rigour within the qualitative research paradigm. Trustworthiness means researchers should clearly describe amongst others : what was done by whom during the inductive, comparative analytical process and if/how the perspectives of multiple coders were used. Researchers triangulation which has been done in our study is one of the strategies to enhance trustworthiness.  In a more quantitative research paradigm trustworthiness is indeed the analogue for reliability. Because this paper was written within a qualitative research paradigm we choose to use the corresponding terminology.

Refs:

- Kornbluh, M. Combatting challenges to establishing trustworthiness in qualitative research. Qualitative Research in Psychology 2015, 12, 397-414.

- Korstjens, I.; Moser, A. Series: practical guidance to qualitative research. Part 4: trustworthiness and publishing. European Journal of General Practice 2018, 24, 120-124.

-Kuper, A.; Reeves, S.; Levinson, W. An introduction to reading and appraising qualitative research. Bmj 2008, 337, a288.

Did the GPs have any training on AB prescribing (beyond medical school) or were they given any materials on AB prescribing at any point in time?

Thank you for this question. There are guidelines, online learning modules, individual prescribing feedback, yearly (public) campaigns with patient information materials, etc.. available for all Belgian GPs. But it depends on an individual level if GPs came in contact with these, followed training, etc… On OOH level there was no intervention, materials or feedback yet at the time of the video-recordings.

Findings are interesting but it would be useful to have some statistics (especially within Theme 1) to be able to learn more concretely from them.  Even just “x was more likely to lead to y”. It would also be useful to have some numbers to understand where the thresholds lie. For example, how long does the expected wait to see a routine (or follow-up) GP for the OOH GP to feel that she/he should prescribe ABs?

*We understand your questions. However this paper was written within a qualitative research paradigm. We used a interpretivist/constructivist epistemological approach whereby we searched for interpretation and meaning in our data. Our research focuses on the questions “why?” and “how?”. Collection and analysis of data was done within this chosen paradigm and was not setup to use in a quantitative approach. For example questions were not asked in a standardized way to all participants. We do not know the potential answers from participants who did just not mention a certain issue. Numbers dependent on sample size, and sample size was not chosen for this purpose. Themes were not only chosen because of their recurrence within interviews, but also for their relevance for our research question and/or their innovativeness. Quotes from the data are supplied as supplementary material to show some examples to illustrate the variety and richness within the themes. From the data collected it’s not possible to answer the question: “how long does the expected wait to see a routine (or follow-up) GP for the OOH GP to feel that she/he should prescribe ABs?”

 Refs:

-Malterud, K. Qualitative research: standards, challenges, and guidelines. The lancet 2001, 358, 483-488.

-Malterud, K. The art and science of clinical knowledge: evidence beyond measures and numbers. The Lancet 2001, 358, 397-400.

- Kuper, A.; Reeves, S.; Levinson, W. An introduction to reading and appraising qualitative research. Bmj 2008, 337, a288.

English needs some work, especially in the Abstract.

A few English mistakes in the main text:

Word missing: Video recording of primary care consultations is a valuable

“A video of their own consultation was used as a prompt to explore the tacit knowledge (that what they know but find difficult to describe)[16], beliefs, attitudes, social influences, communication… that drove their behaviour.“

“beliefs-in action” (there should be no hyphen)

“All about 175 GPs of the region have the obligation to participate in the OOH care system.”

*Thank you. We corrected this.

“If this happens… than…” (it should be “then”)

*Thank you for this remark. Unfortunately we were not able to find the sentence where we misused the word than. We will keep specific attention on this when editing the final text.